# Vitamin D Receptor Gene Polymorphism and Vitamin D Status in Population of Patients with Cardiovascular Disease—A Preliminary Study

**DOI:** 10.3390/nu13093117

**Published:** 2021-09-06

**Authors:** Mohamed Abouzid, Marlena Kruszyna, Paweł Burchardt, Łukasz Kruszyna, Franciszek K. Główka, Marta Karaźniewicz-Łada

**Affiliations:** 1Department of Physical Pharmacy and Pharmacokinetics, Poznan University of Medical Sciences, 6 Święcickiego Street, 60-781 Poznan, Poland; mabouzid@outlook.com (M.A.); glowka@ump.edu.pl (F.K.G.); 2Department of Hypertension, Angiology, and Internal Medicine, Poznan University of Medical Sciences, Długa ½, 60-848 Poznan, Poland; mkruszyna@ump.edu.pl (M.K.); pburchardt@ump.edu.pl (P.B.); 3Department of Cardiology, J. Struś Hospital, Szwajcarska 3, 61-285 Poznan, Poland; 4Department of Vascular and Endovascular Surgery, Angiology and Phlebology, Poznan University of Medical Sciences, Długa ½, 60-848 Poznan, Poland; lukaszkruszyna@ump.edu.pl

**Keywords:** *ApaI*, *BsmI*, *FokI*, *TaqI*, hypertension, obesity, 25-hydroxyvitamin D, epi-25-hydroxyvitamin D

## Abstract

The association between vitamin D receptor (VDR) polymorphism and the risk of cardiovascular diseases (CVD) remains unclear. This study aimed to assess a relationship between the *VDR* genotypes, plasma concentrations of vitamin D metabolites, and the occurrence of cardiovascular and metabolic disorders. Fifty-eight patients treated for various cardiological afflictions were included. Identification of *VDR* polymorphisms: *ApaI*, *TaqI*, *BsmI*, and *FokI* were carried out using the PCR-RFLP method. Plasma concentrations of 25-hydroxyvitamin-D2, 25-hydroxyvitamin-D3, and 3-epi-25-hydroxyvitamin D3 were assessed by the UPLC-MS/MS method. Lower incidence of *BsmI* AA genotype in the studied patients was observed compared with healthy controls, but the difference was insignificant. Among patients with the TT genotype, frequency of hypertension was higher than among carriers of other *ApaI* genotypes (*p* < 0.01). In addition, carriers of the TT *ApaI*, TC *TaqI*, and GA *BsmI* genotypes had an increased risk of obesity, while the presence of the *FokI* TT genotype was associated with a higher incidence of heart failure and hypertension. In conclusion, the *BsmI* AA genotype can be protective against CVD, but this observation needs study on a larger group of patients. Particular *VDR* genotypes were associated with 25-hydroxyvitamin-D levels, and the mechanism of this association should be further investigated.

## 1. Introduction

Cardiovascular diseases (CVDs) are the leading cause of death in the world and include heart and vessel disease related to a process of atherosclerosis [1]. CVDs refer to the following entities: coronary artery disease (CAD), cerebrovascular disease, peripheral artery disease (PAD), and aortic atherosclerosis [2]. The etiology of CVD is multifactorial and involves a complex interaction of political, social, behavioral, physical, biological, and genetic factors [3]. Over 100 single nucleotide polymorphisms (SNPs) associated with CVD have been identified [4]. However, the precise mechanism of how SNPs influence CVD remains poorly recognized [5].

The endocrine vitamin D (vitD) system involves a wide range of biological processes, including cell proliferation, control and differentiation, bone synthesis, and immune response modulation [6]. The mechanism of vitD involves binding the active metabolite calcitriol to the vitamin D receptor (VDR), which belongs to the steroid hormone family of nuclear receptors accountable for the transcriptional regulation of several hormone-responsive genes [7]. Modern studies show that there are four well-characterized *VDR* polymorphisms; *FokI*, *BsmI*, *ApaI*, and *TaqI* (Figure 1), in addition to Cdx2, poly (A), A-1012G, and Tru 91, which are still rarely analyzed [8]. *VDR* gene polymorphisms may potentially affect CVD due to the presence of VDRs in all main cardiovascular cell types such as vascular smooth muscle cells (VSMC) [9], endothelial cells (ECs) [10], cardiomyocytes [11], platelets [12], and most immune cells [13]. Moreover, the *VDR* is a major element in regulating the expression of varied proteins involved in regulating the cardiovascular system, such as renin [14], endothelial nitric oxide synthase [15], and NADPH oxidase (NOX) [16].

Investigating the association of genetic variation in the VDR with CVD may therefore provide further insights regarding the influence of the endocrine vitD system on the incidence rate or severity of the disease. To date, only a few studies have assessed the association between vitD related *ApaI, BsmI*, *FokI,* and *TaqI* SNPs and the occurrence of CVD, but their results are inconsistent. Ortlepp et al. [17] declared no association between *BsmI* polymorphism and CAD in the European population, while van Schooten et al. [18] obtained contradictory results. In addition, *VDR FokI* polymorphisms appear to be associated with CAD in Han Chinese adults [19]. There is only one study on *BsmI* and *FokI* polymorphisms and CVD in the Polish population [20]. No studies were conducted to evaluate all aforementioned polymorphisms and 25-hydroxy metabolites of vitD in Polish patients with CVD.

Evidence indicates that measuring the vitD status in the body may be beneficial to predict patients’ susceptibility to CAD [21,22]. Vitamin D2 (ergocalciferol) and vitamin D3 (cholecalciferol) are the main functionally inactive precursors for vitD [23]. Upon hydroxylation in the liver, they yield 25-hydroxyvitamin-D2 (25(OH)D2) and 25-hydroxyvitamin-D3 (25(OH)D3), which are used to determine vitD status [24]. Additional epimerization by 3-epimerase produces 3-epi-25-hydroxyvitamin D2 and 3-epi-25-hydroxyvitamin D3 (3-epi-25(OH)D2, 3-epi-25(OH)D3) [25]. The biological activity of C3 epimeric metabolites has been discussed widely in in vitro models; however, there are few research articles about the in vivo model [26]. According to a recent study, it is believed that there are differences between genetic determinants of non-C3-epimers and C3-epimers [27]. The physiological role of 3-epi-25(OH)D is undefined, but its concentration may influence the clinical interpretation of vitamin D status. According to Strathmann [28], 3% of adults would be misclassified as sufficient if 3-epi-25(OH)D3 was included in the quantification of 25(OH)D3. Moreover, recent research shows that the presence of the C-3 epimers may constitute up to 50% of the 25(OH)D content in adults [29]. Because the clinical impact of 3-epi-25(OH)D3 remains ambiguous, many studies are trying to measure it in various situations such as in pregnancy [30], chronic liver disease [31], and thyroid disorders [32]. Despite Stokes et al. [31] not finding a significant relationship in 3-epi-25(OH)D3 levels in some diseases, Bennett et al. reported [30] that 3-epi-25(OH)D3 levels are key markers for glycemic control in women with type 1 diabetes.

Although the relationship between *VDR* polymorphism and 25(OH)D levels was confirmed, the mechanism of this linkage is unknown [33]. Divanoglou et al. [34] suggested that epigenetic modifications of *VDR* regulate the conversion of vitD into its metabolites through CYP450 and thus affect 25(OH)D concentrations.

Therefore, this study was undertaken to investigate the association between four SNPs of *BsmI* (rs1544410), *ApaI* (rs7975232), *TaqI* (rs731236), and *FokI* (rs2228570), plasma concentrations of 25-hydroxy metabolites of vitD, including 25(OH)D2, 25(OH)D3 and 3-epi-25(OH)D3, and cardiovascular and metabolic disorders in Polish patients.

## 2. Materials and Methods

### 2.1. Patient Selection

The participants of this study were 58 patients (49 males and 9 females) with CVD admitted to the hospital for planned angioplasty/angiography due to CAD (52 patients) or PAD (6 patients). The average age of the patients was 62.9 ± 8.2 years and body mass index (BMI) 28.6 ± 3.8 kg/m^2^. Exclusion criteria included: supplementation with vitD, acute myocardial infarction, malignancies, impaired renal function determined by serum creatinine concentration >2 mg/dL, and current liver dysfunction. A total of 90 samples were collected from the patients in different seasons, autumn–winter and spring–summer (Table 1). The study design and conduction are within agreement with the outlined ethical principles in the Declaration of Helsinki. The study protocols (no. 273/15, and 58/20) were accepted and approved by the Local Ethical Committee of Poznan University of Medical Sciences (PUMS). Written informed consent was obtained from each participant. In addition, 25(OH)D2, 25(OH)D3 and 3-epi-25(OH)D3 concentrations were assessed using the validated UPLC-MS/MS method [29]. Briefly, 200 μL of patients’ plasma was mixed with 20 μL of methanol and 20 μL of the internal standard, D6-25-hydroxyvitamin D3. The analytes were extracted with two portions of 1000 μL of hexane. After evaporation, the residue was dissolved in 200 μL of methanol-water (80:20, *v*/*v*) solution, and 10 μL was injected directly into the UPLC Nexera coupled to a triple quadrupole mass spectrometer LCMS-8030 (Shimadzu, Kyoto, Japan). The separation of 25(OH)D2, 25(OH)D3 and 3-epi-25(OH)D3 was performed in the Kinetex 2.6μm F5 analytical column (50 mm×2.1 mm) (Phenomenex, Torrance, CA, USA) with a mobile phase composed of methanol–water (80:20, *v*/*v*) containing formic acid (0.1%, *v*/*v*). Mass spectrometry detection of the analytes was performed in a positive electrospray ionization mode. The UPLC-MS/MS method was linear in the concentration range of 1–100 ng/mL. Recovery of the analytes from plasma samples were 85–107%. The inter- and inter-day precision was in the range of 4–18%, while accuracy of the method was 86–115% [29]. The applied UPLC-MS/MS method enabled for separating 25(OH)D2 and 25(OH)D3 from 3-epi-25(OH)D3, which allowed for avoiding overestimation of the 25(OH)D3 concentration in the studied group of patients with CVD.

### 2.2. Genetic Analyses

The isolation of the genomic DNA from 200 μL of blood was performed using a commercially available kit according to the manufacturer’s instruction (GenMATRIX Quick Blood Purification Kit, EURx, Poland). To identify *ApaI*, *TaqI*, *BsmI*, and *FokI* polymorphisms, polymerase chain reaction (PCR) coupled with restriction fragment length analysis (RFLP) was performed. Primers were used as reported by Pani et al. [35]. Their sequences were checked by the Primer3Plus software [36]. The primers’ sequences, their annealing temperatures, the length of the products obtained, and the results of restriction were presented in the supplementary material (Appendix A).

### 2.3. Statistical Analysis

Statistical analysis was performed using TIBCO Statistica™ 13.4.0 with Medical Bundle (StatSoft Inc., Tulsa, OK, USA). Descriptive statistics were used to summarize the continuous variables for age, weight, height and BMI, lipid profiles, and 25(OH)D2, 25(OH)D3, and 3-epi-25(OH)D3 concentrations. Descriptive statistics are presented for the normal variables as mean and standard deviation (SD), and for non-normal variables as the median and interquartile range (IQR). Deviation from the normality of the distribution of vitD concentrations was verified with the Shapiro–Wilk test. Chi-squared test (χ^2^) goodness-of-fit was used to test Hardy–Weinberg equilibrium for detecting the allele frequency. The Kruskal–Wallis ANOVA by Ranks, followed by Dunn–Bonferroni post hoc, was applied to compare the differences between multiple groups. A *p*-value < 0.05 was considered statistically significant for all tests. In the next step, we used multivariate logistic regression analysis to find parameters most strongly and independently related to the occurrence of patients’ medical conditions such as diabetes, hypercholesterolemia, hypertension, and heart failure.

## 3. Results

The distribution of *BsmI*, *TaqI*, and *FokI* genotypes in the studied group was consistent with Hardy–Weinberg equilibrium (χ^2^ < 3.84, α = 0.05). Slight deviations were observed for *ApaI* genotypes (χ^2^ = 4.08, α = 0.05). Allele frequencies of the polymorphisms in the studied population and healthy Polish subjects are presented in Table 2. The comparison shows that these frequencies are similar. The biggest difference concerns the *BsmI* polymorphism, where the prevalence of the AA genotype is almost two times lower than in healthy subjects.

The contribution of the *ApaI*, *TaqI*, *BsmI*, and *FokI* genetic variants to the variability in demographic, biochemical, and medical conditions were assessed. We also analyzed the concentrations of the 25(OH)D2, 25(OH)D3, and 3-epi-25(OH)D3 levels in relation to the studied polymorphisms and different seasons.

### 3.1. ApaI

When analyzing *ApaI* polymorphism, significant differences in frequency (*p* = 0.01) were found in the occurrence of hypertension in the studied group. Among people with the GG genotype, 71% had hypertension, 48% had the disease in GT genotype, whereas there were 93% patients with the TT genotype. Additionally, a higher incidence of obesity was observed in carriers of at least one T allele, and the result has hovered around significance (*p* = 0.06). Patients with GG genotype had significantly (*p* < 0.01) higher plasma levels of 25(OH)D3 and 3-epi-25(OH)D3 compared with the GT carriers (Table 3).

### 3.2. TaqI

In patients with different *TaqI* genotypes, significant differences between the values were found in BMI, and the incidence of obesity and hypercholesterolemia. Obesity (*p* < 0.01) was noticed in 22% of patients with the CC genotype, 50% of patients with the TC genotype, and 12% patients with the TT genotype. Hypercholesterolemia (*p* = 0.05) was found in 67% carriers of genotype CC, 45% with the TC genotype, 69% with the TT genotype. 25(OH)D2 levels were lower in TT genotype than CC and TC genotypes and results were a reliable trend (*p* = 0.06). In addition, 3-epi-25(OH)D3 levels were significantly lower in TC genotype compared with CC genotype (*p* = 0.03) (Table 4).

### 3.3. BsmI

Significant differences (*p* = 0.04) in frequency were found in the prevalence of obesity in people with different *BsmI* genotypes. Among people with the GG genotype, 10% were obese, 41% of people with the GA genotype had the disease, and with the AA genotype, there were 25% patients. The levels of 25(OH)D3 and 3-epi-25(OH)D3 were significantly (*p* = 0.01) higher in GG genotype compared with GA (Table 5).

### 3.4. FokI

Significant differences (*p* = 0.03) in frequency were found in the occurrence of heart failure in people with different *FokI* genotypes. Among people from heart failure was diagnosed in 40% of patients with the TT genotype and 13% of patients with the CC genotype with no heart failure occurrences in TC genotype. Additionally, a higher incidence of hypertension was observed in people with at least one T allele, with all patients with the TT genotype having the disease. However, the differences in the incidence of hypertension in people with different *FokI* genotypes were at the borderline of statistical significance (*p* = 0.05). Levels of 3-epi-25(OH)D3 were significantly higher in TC genotype compared with CC genotype (*p* = 0.03) (Table 6).

### 3.5. Multifactorial Model

Backward stepwise regression shows that hypertension can be predicted by BMI, HbA1C [%], and *ApaI* GT (Table 7). HbA1C [%] was a 32-fold risk factor of hypertension (OR = 32.39, 95%CI = 0.17–6.79, *p* = 0.04), while BMI (OR = 2.11, 95%CI = 0.08–1.41, *p* = 0.03), and *ApaI* GT (OR = 0.05, 95%CI = −5.92-(−0.02), *p* = 0.03). *ApaI* GT genotype was protective.

### 3.6. Seasonal Analysis

Inter-seasonal changes have been analyzed for 25(OH)D2, 25(OH)D3, and 3-epi-25(OH)D3 and *VDR* polymorphisms. Only 25(OH)D3 levels were found to vary significantly according to the season in some *VDR* polymorphisms (Table 8). 25(OH)D3 concentrations were significantly higher in the spring–summer period than autumn–winter in several genotypes: *ApaI* GT (*p* = 0.02), *FokI* TC (*p* = 0.04), *FokI* CC (*p* = 0.02), and *BsmI* GA (*p* < 0.01). Although *TaqI* TT genotype was sensitive to 25(OH)D3 seasonal variation, the differences were insignificant. Intra-seasonal 25(OH)D3 levels were significantly the highest in *ApaI* GG compared with *ApaI* GT and TT (*p* = 0.01, *p* < 0.01, respectively). In addition, 25(OH)D3 was significantly (*p* < 0.01) higher in *BsmI* GG compared with *BsmI* GA. Seasonal variation data associated with 25(OH)D2 and 3-epi-25(OH)D3 are not shown.

## 4. Discussion

This study aimed to examine the four major *VDR* polymorphisms: *ApaI, TaqI, BsmI*, and *FokI*, plasma concentrations of 25-hydroxy metabolites, and occurrence of such conditions as hypertension, diabetes, obesity, hypercholesterolemia in Polish patients with CVD. Allele frequencies observed in the studied Polish population were similar to those reported by Harbuzova et al. [38] and Ortlepp et al. [17] in Ukrainian and German patients. In addition, frequencies of *VDR* polymorphisms were compared with healthy volunteers in Poland [37] and similar results were observed. Only the *BsmI* AA genotype was almost twice lower in CVD patients than healthy volunteers; however, the difference was not statistically significant (Table 2). A meta-analysis conducted by Zhu et al. [39] revealed a lower *BsmI* AA genotype frequency in hypertension patients than healthy controls. The authors suggested that the population carrying the *VDR BsmI* AA genotype has a lower hypertension risk than those carrying the GA or GG genotype [39]. Therefore, AA genotype can be protective against CVD, but a study on a larger population is needed.

### 4.1. ApaI

Patients with GT genotype had a lower frequency of hypertension than GG and TT genotypes. Despite the noted higher incidence of obesity with at least one T allele, the result has hovered around significance. There were no observed significant relationships between the presence of *ApaI* genotypes and diabetes, being overweight, obesity, hypercholesterolemia, or heart failure. According to Alizadeh et al. [40], *ApaI* polymorphism was not associated with CAD. However, Lu et al. [41] mentioned that a decreased risk of CAD was associated with the *ApaI* heterozygote genotype and an increased risk with the recessive genotype but only in patients with type 2 diabetes (T2D). In addition, according to Ferrarezi et al. [42], AAC haplotype (*BsmI/ApaI/TaqI*) was significantly associated with CAD in T2D patients.

In our study, the multivariate analysis revealed that BMI, HbA1C [%], and *ApaI* GT genotype were significantly linked to hypertension (Table 7). Shen et al. [43] observed that people with the *ApaI* GT genotype had higher cholesterol and LDL levels and the presence of the T allele was associated with triceps skinfold thickness and body fat percentage in the Chinese population. In our study, the concentration of total cholesterol, HDL, LDL, triglycerides, HbA1C, and creatinine did not differ significantly according to genotype (Table 3). However, patients with *ApaI* GG genotype had significantly higher plasma levels of 25(OH)D3 and 3-epi-(OH)D3 compared with the *ApaI* GT carriers. Maaty et al. [44] confirmed that *ApaI* GG genotype was associated with greater levels of 25(OH)D in patients with CAD in Egypt. Another study on Indian patients with angiography by Shanker et al. [45] revealed that *VDR* genotypes did not show any association with either vitD levels or CAD.

### 4.2. TaqI

Obesity was the most prevalent among patients with TC genotype, and they also had the highest BMI. In addition, patients with TT genotype had a significantly higher frequency of hypercholesterolemia. In our CVD patients, being overweight was a 6-fold risk factor for hypercholesterolemia regardless of the polymorphism type (OR = 6.4, 95%CI = 1.0–40.3, *p* = 0.05). The association of obesity with the *TaqI* genotype was also confirmed in other studies [46,47]. Al-Hazmi et al. [46] concluded that CC genotype was significantly associated with a higher obesity risk in Saudi men. Similarly, Al-Daghri et al. [47] mentioned that GTA [(rs731236 (G), rs1544410 (T), and rs7975232 (A)] haplotype was significantly associated with the presence of obesity and higher BMI. Moreover, Lu et al. [41] observed that the presence of the *TaqI* restriction site might be associated with an increased risk of developing CVD, especially ischemic heart disease. Therefore, modifying therapeutic management is necessary, especially with patients with increased BMI and cholesterol levels. Analysis of 25(OH)D metabolites revealed that 25(OH)D2 levels were lower in *TaqI* TT genotype compared with *TaqI* CC and *TaqI* TC genotypes, and the result was a reliable trend. In addition, 3-epi-25(OH)D3 levels were significantly lower in the *TaqI* TC genotype compared with *TaqI* CC genotype. In contrast, Maaty et al. [44] reported that *TaqI* polymorphism was not found to predict 25(OH)D2 and 25(OH)D3 levels.

### 4.3. BsmI

A significant relationship between genotype and obesity incidence was also observed in the case of *BsmI* polymorphism. The greatest number of cases of obesity was found in patients with GA genotype, followed by AA, while the lowest obesity cases were found in patients with GG genotype (Table 5). However, a contradicted result has been reported by Hazmi et al. [46] since, in Saudi men, GG genotype was associated with higher obesity risk. In contrast, Laczmanski et al. [20] mentioned that *BsmI* common allele A is associated with biochemical risk factors (cholesterol, triglycerides, HDL, LDL, glucose, insulin, homeostatic model) of CVD in older Polish Caucasian men and women population. We did not record a significant association between hypertension and *BsmI* polymorphism. In addition, Ortlepp et al. [17] reported that *BsmI* polymorphism is insignificant with the incidence and severity of CAD. However, according to Zhu et al. [39], patients with the AA genotype had a lower incidence of hypertension compared with other genotypes. In addition, Lee et al. [48] observed that arterial hypertension occurs more often in people with at least one G allele (OR = 2.1; 95%CI = 1.0–4.4; *p* = 0.05). Moreover, systolic and diastolic blood pressure were higher (by 2.7–3.7 mm Hg and 1.9–2.5 mm Hg, respectively) than in the AA genotype carriers [48]. Finally, Rahmadhani et al. [49] found that AA genotype in adolescents was associated significantly with higher insulin resistance but not with obesity. The analysis of 25(OH)D levels revealed that 25(OH)D3 and 3-epi-25(OH)D3 were significantly higher in *BsmI* GG genotype compared with *BsmI* GA (Table 5). However, according to Cobayashi et al. [50], mutant allele T was negatively associated with serum 25(OH)D.

### 4.4. FokI

Significant differences were found in the occurrence of heart failure in people with different *FokI* genotypes. Heart failure was diagnosed in 40% of patients with the TT genotype and 13% of patients with the CC genotype with no heart failure occurrences in TC genotype (Table 6). Hao et al. [51] stated that the T allele was more common in Chinese patients with heart failure than in healthy volunteers (OR = 2.45, *p* < 0.01). In addition, Lu et al. [41] mentioned that TC may play a protective role in CAD. Contradicted results were reported by Pan et al. [52] who mentioned that Chinese patients with CAD show no significant difference between *BsmI* and *FokI* polymorphism allele frequencies and CAD. Despite this fact, we noticed a higher incidence of hypertension in people with at least one T allele, and all patients with the TT genotype had the disease. The results were at the borderline of statistical significance (Table 6). According to Swapna et al. [53], CC genotype and allele T were at a significantly greater risk for developing hypertension in Indian patients. A study by Schuch et al. [54] confirmed the association between TC polymorphism with the development of higher insulin resistance, cholesterol and triglycerides, and lower HDL levels. A significant association was observed between levels of 3-epi-25(OH)D3 and the presence of *FokI* TC genotype compared with *FokI* CC genotype (Table 6). Biswas et al. [55] reported a significant relation between *FokI* TT genotype polymorphism and increased level of serum 1, 25(OH)D3 in autism spectrum disorder patients.

### 4.5. Seasonal Analysis

Various studies have mentioned seasonal variation in vitD [56,57,58,59,60]. Some studies have shown lower levels of serum 25(OH)D in winter, whereas, in others, the prevalence of vitD deficiency did not differ by seasonal changes and remained stable even in a sunny climate [57,58,59,60,61,62]. In our sample, we have noticed significant changes in 25(OH)D metabolites between seasons according to *VDR* polymorphisms. Concerning inter-seasonal changes in the whole group of patients, 25(OH)D3 levels were higher in spring–summer compared with autumn–winter (*p* < 0.01). In-depth analysis shows that the four genotypes—*ApaI* GT, *FokI* TC, *BsmI* GA, and *FokI* CC—were significantly sensitive to seasonal changes (Table 8).

## 5. Conclusions

Lower frequency of *BsmI* AA genotype in the studied patients compared with healthy controls suggests that the population carrying *VDR BsmI* AA genotype has a lower risk of CVD. However, this observation needs further investigation on a larger group of patients. Since particular *ApaI*, *TaqI*, *BsmI,* and *FokI* genotypes seem to predict 25(OH)D levels, the mechanism of this association should be studied. A triangular relationship between the *VDR* polymorphisms, disease incidence, and 25(OH)D metabolites may be helpful in the prediction of the CVD risk.

## Figures and Tables

**Figure 1 nutrients-13-03117-f001:**
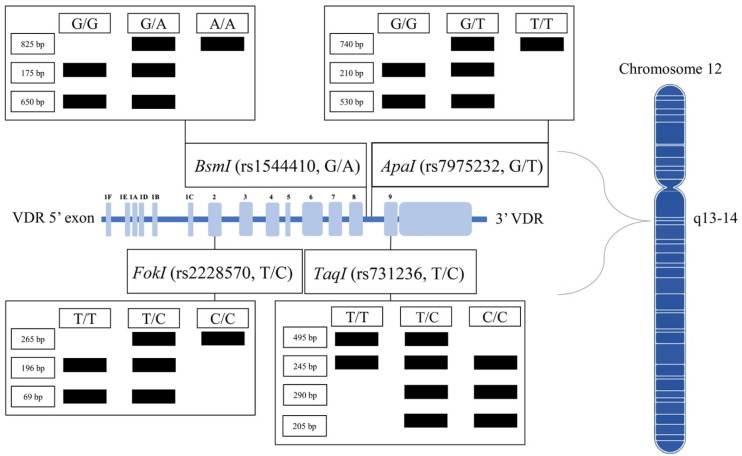
Common polymorphisms in *VDR* genes and its PCR-RFLP based genotyping.

**Table 1 nutrients-13-03117-t001:** Characteristics of patients. Values are given as the number of patients N (%) or mean ± SD.

Characteristic	N (%)
Sex & demographic data:	
Male	49 (84)
Female	9 (16)
Age [years]	62.91 ± 8.21
Weight [kg]	83.82 ± 14.80
Height [m]	1.71 ± 0.85
BMI [kg/m^2^]	28.66 ± 3.83
Diagnosis & medical history:	
Overweight	44 (76)
Obesity	16 (28)
Diabetes	21 (36)
Hypercholesterolemia	34 (59)
Hypertension	41 (71)
Heart failure	6 (10)
Stroke	3 (5)

**Table 2 nutrients-13-03117-t002:** Distribution of genotypes (N, (%)).

	Patients with Cardiovascular Disease (N = 58)	Healthy Subjects (N = 142, [37])	*p*
*ApaI*			
GG	22 (37.9)	57 (40.1)	0.77
GT	21 (36.2)	49 (34.5)	0.82
TT	15 (25.9)	36 (25.4)	0.94
*FokI*			
TT	6 (10.3)	17 (12.0)	0.74
TC	21 (36.2)	42 (29.6)	0.36
CC	31 (53.4)	83 (58.5)	0.52
*TaqI **			
TT	26 (45.6)	69 (48.6)	0.95
TC	22 (38.6)	49 (34.5)	0.65
CC	9 (15.8)	24 (16.9)	0.63
*BsmI*			
GG	21 (36.2)	49 (34.5)	0.82
GA	29 (50)	59 (41.5)	0.19
AA	8 (13.8)	34 (23.9)	0.11

* missing results for one patient.

**Table 3 nutrients-13-03117-t003:** Patient characteristics according to the *ApaI* polymorphism.

Variable	GG (N = 22)	GT (N = 21)	TT (N = 15)	*p*
Weight [kg] ^¶^	81.45 ± 9.88	82.35 ± 14.48	90.69 ± 20.08	0.38
BMI [kg/m^2^] ^¶^	27.34 ± 2.68	28.45 ± 3.54	30.95 ± 4.80	0.08
Cholesterol [mmol/l] ^¶^	4.23 ± 1.09	4.31 ± 1.36	4.27 ± 1.08	0.89
HDL [mmol/l] ^¶^	1.31 ± 0.34	1.11 ± 0.37	1.29 ± 0.28	0.12
LDL [mmol/l] ^¶^	2.37 ± 0.95	2.60 ± 1.32	2.38 ± 0.92	0.93
Triglycerides [mmol/l] ^¶^	1.31 ± 0.66	1.35 ± 0.45	1.32 ± 0.84	0.68
HbA1C [%] ^¶^	5.84 ± 1.06	5.78 ± 0.72	6.49 ± 1.56	0.32
Creatinine [μmol/l] ^¶^	89.79 ± 33.88	79.11 ± 19.42	84.62 ± 14.63	0.33
25(OH)D2 [ng/mL] ^§^	1.74 (1.10–2.93)	1.55 (1.07–2.34)	1.34 (0.18–2.62)	0.48
25(OH)D3 [ng/mL] ^§^	21.12 (13.44–24.62)	8.33 (3.46–16.66)	12.25 (4.75–14.95)	<0.01 ^α^^$^
3-epi-25(OH)D3 [ng/mL] ^§^	3.00 (2.33–3.78)	1.80 (1.04–2.79)	1.85 (1.16–3.21)	0.03 ^α^^$^
Diabetes ^β^	7 (32%)	7 (33%)	7 (47%)	0.69
Overweight ^β^	18 (82%)	14 (66%)	12 (80%)	0.67
Obesity ^β^	3 (14%)	6 (29%)	7 (47%)	0.06
Hypertension ^β^	17 (77%)	10 (48%)	14 (93%)	<0.01 ^α^^¥^
Hypercholesterolemia ^β^	17 (77%)	9 (43%)	8 (53%)	0.07
Heart failure ^β^	2 (9%)	4 (19%)	0 (0%)	0.16

^¶^—Data are presented as mean ± SD; ^§^—Data are presented as median (IQR)**,**
^β^—Data are presented as N (%); ^α^—Significant at *p* < 0.05; ^$^—Kruskal–Wallis ANOVA; ^¥^—Pearson’s & M-L Chi-square. Abbreviations: HDL—high density lipoprotein; LDL—low density lipoprotein; HbA1C — hemoglobin A1c.

**Table 4 nutrients-13-03117-t004:** Patient characteristics according to the *TaqI* polymorphism.

Variable	CC (N = 9)	TC (N = 22)	TT (N = 26)	*p*
Weight [kg] ^¶^	79.22 ± 15.49	91.17 ± 17.90	80.73 ± 9.86	0.05
BMI [kg/m^2^] ^¶^	28.14 ± 3.25	30.54 ± 4.76	30.16 (28.34–33.93) ^§^	0.04 ^α^^$^
Cholesterol [mmol/l] ^¶^	4.27 ± 1.02	4.40 ± 1.28	4.07 ± 1.10	0.42
HDL [mmol/l] ^¶^	1.29 ± 0.31	1.16 ± 0.34	1.29 ± 0.37	0.39
LDL [mmol/l] ^¶^	2.46 ± 0.84	2.59 ± 1.23	2.25 ± 0.96	0.57
Triglycerides [mmol/l] ^¶^	1.18 ± 0.52	1.48 ± 0.66	1.25 ± 0.68	0.28
HbA1C [%] ^¶^	6.31 ± 1.54	5.95 ± 1.01	5.87 ± 1.04	0.64
Creatinine [μmol/l] ^¶^	86.44 ± 15.26	80.90 ± 19.13	88.40 ± 33.20	0.49
25(OH)D2 [ng/mL] ^§^	1.81 (1.15–2.98)	1.74 (1.34–3.68)	0.56 (0.18–1.35)	0.06
25(OH)D3 [ng/mL] ^§^	17.99 (4.28–24.07)	8.81 (3.46–14.89)	12.78 (6.22–25.03)	0.04 ^α^^$^
3-epi-25(OH)D3 [ng/mL] ^§^	3.03 (1.79–3.61)	1.69 (0.96–2.86)	2.41 (1.54–3.17)	0.03 ^α^^$^
Diabetes ^β^	4 (40)	8 (36)	8 (31)	0.84
Overweight ^β^	8 (80)	15 (68)	20 (77)	0.76
Obesity ^β^	2 (20)	11 (50)	3 (12)	<0.01 ^α^^¥^
Hypertension ^β^	8 (80)	13 (59)	20 (77)	0.09
Hypercholesterolemia ^β^	6 (67)	10 (45)	18 (69)	0.05 ^α^^¥^
Heart failure ^β^	0 (0)	4 (18)	2 (8)	0.28

^¶^—Data are presented as mean ± SD; ^§^—Data are presented as median (IQR); ^β^—Data are presented as N (%); ^α^—Significant at *p* < 0.05; ^$^—Kruskal–Wallis ANOVA; ^¥^—Pearson’s & M-L Chi-square.

**Table 5 nutrients-13-03117-t005:** Patient characteristics according to the *BsmI* polymorphism.

Variable	GG (N = 21)	GA (N = 29)	AA (N = 8)	*p*
Weight [kg] ^¶^	80.56 ± 10.16	87.60 ± 16.65	81.14 ± 17.33	0.35
BMI [kg/m^2^] ^¶^	27.23 ± 2.67	29.65 ± 4.40	28.80 ± 3.40	0.19
Cholesterol [mmol/l] ^¶^	4.34 ± 1.10	4.23 ± 1.31	4.22 ± 0.86	0.93
HDL [mmol/l] ^¶^	1.31 ± 0.37	1.14 ± 0.31	1.40 ± 0.31	0.07
LDL [mmol/l] ^¶^	2.52 ± 0.90	2.46 ± 1.24	2.30 ± 0.93	0.75
Triglycerides [mmol/l] ^¶^	1.25 ± 0.71	1.43 ± 0.63	1.14 ± 0.50	0.29
HbA1C [%] ^¶^	5.93 ± 1.06	5.87 ± 0.94	6.52 ± 1.77	0.57
Creatinine [μmol/l] ^¶^	90.56 ± 37.15	81.07 ± 16.79	84.57 ± 17.10	0.70
25(OH)D2 [ng/mL] ^§^	1.7 (1.09–2.98)	1.71 (0.98–2.78)	1.32 (0.18–1.72)	0.59
25(OH)D3 [ng/mL] ^§^	21.39 (12.15–24.21)	10.98 (3.46–16.59)	11.10 (4.75–25.03)	0.01 ^α^^$^
3-epi-25(OH)D3 [ng/mL] ^§^	2.89 (2.33–3.98)	1.81 (1.04–2.94)	1.91 (1.31–3.21)	0.01 ^α^^$^
Diabetes ^β^	6 (29)	11 (38)	4 (50)	0.66
Overweight ^β^	15 (71)	22 (76)	7 (88)	0.52
Obesity ^β^	2 (10)	12 (41)	2 (25)	0.04 ^α^^¥^
Hypertension ^β^	14 (67)	19 (66)	8 (100)	0.15
Hypercholesterolemia ^β^	13 (61)	17 (59)	4 (50)	0.55
Heart failure ^β^	3 (14)	3 (10)	0 (0)	0.47

^¶^—Data are presented as mean ± SD; ^§^—Data are presented as median (IQR); ^β^—Data are presented as N (%); ^α^—Significant at *p* < 0.05; ^$^—Kruskal–Wallis ANOVA; ^¥^—Pearson’s & M-L Chi-square.

**Table 6 nutrients-13-03117-t006:** Patient characteristics according to the *FokI* polymorphism.

Variable	TT (N = 6)	TC (N = 21)	CC (N = 31)	*p*
Weight [kg] ^¶^	82.00 ± 17.87	84.20 ± 13.94	83.92 ± 15.55	0.84
BMI [kg/m^2^] ^¶^	28.87 ± 2.92	28.93 ± 4.53	28.24 ± 3.46	0.91
Cholesterol [mmol/l] ^¶^	3.88 ± 0.78	4.27 ± 1.14	4.37 ± 1.26	0.62
HDL [mmol/l] ^¶^	1.22 ± 0.44	1.23 ± 0.29	1.24 ± 0.38	0.97
LDL [mmol/l] ^¶^	2.26 ± 0.86	2.42 ± 0.94	2.55 ± 1.21	0.10
Triglycerides [mmol/l] ^¶^	0.92 ± 0.42	1.48 ± 0.80	1.30 ± 0.54	0.22
HbA1C [%] ^¶^	5.45 ± 0.19	5.79 ± 0.54	6.27 ± 1.50	0.41
Creatinine [μmol/l] ^¶^	78.20 ± 14.53	79.77 ± 16.68	90.00 ± 31.18	0.43
25(OH)D2 [ng/mL] ^§^	1.68 (0.25–1.71)	1.62 (1.01–2.93)	1.55 (0.56–2.63)	0.99
25(OH)D3 [ng/mL] ^§^	9.12 (4.52–19.42)	13.67 (6.52–22.67)	14.89 (3.46–21.77)	0.45
3-epi-25(OH)D3 [ng/mL] ^§^	2.44 (1.82–3.09)	3.05 (1.80–3.84)	1.61 (1.04–2.81)	0.03 ^α^^$^
Diabetes ^β^	0 (0)	9 (43)	11 (35)	0.17
Overweight ^β^	5 (83)	17 (81)	21 (68)	0.66
Obesity ^β^	1 (17)	7 (33)	7 (23)	0.79
Hypertension ^β^	5 (83)	17 (81)	18 (58)	0.05 ^α^^¥^
Hypercholesterolemia ^β^	4 (67)	13 (62)	16 (52)	0.78
Heart failure ^β^	2 (33)	0 (0)	4 (13)	0.03 ^α^^¥^

^¶^—Data are presented as mean ± SD; ^§^—Data are presented as median (IQR); ^β^—Data are presented as N (%); ^α^—Significant at *p* < 0.05; ^$^—Kruskal-Wallis ANOVA; ^¥^—Pearson’s & M-L Chi-square.

**Table 7 nutrients-13-03117-t007:** Determinants of hypertension in the studied group of patients with cardiovascular disease.

	Estimate	Standard Error	Wald Stat.	CL 95%		*p*	Odds Ratio	Confidence OR	
				Lower	Upper			−95%	95%
**BMI**	0.75	0.34	4.87	0.08	1.41	0.03	2.11	1.09	4.11
**HbA1C [%]**	3.48	1.69	4.25	0.17	6.79	0.04	32.39	1.19	884.43
***ApaI* GT**	−3.08	1.45	4.49	−5.92	−0.23	0.03	0.05	<0.01	0.79

**Table 8 nutrients-13-03117-t008:** Inter-seasonal changes of 25(OH)D3 in patients’ plasma according to the VDR polymorphism.

	N	Autumn–Winter	Spring–Summer	Z	*p*
** *ApaI* **					
GG	14	21.56 (15.52–24.01) ^├^	17.30 (11.61–27.49)	1.65	0.10
GT	12	4.35 (2.92–8.81)	13.62 (8.39–18.41)	2.29	0.02 ^α^
TT	9	6.22 (3.45–12.15) ^├^	14.95 (12.43–25.03)	1.59	0.11
** *TaqI* **					
TT	8	9.43 (5.16–12.77)	24.65 (13.69–32.53)	1.82	0.07
CT	12	5.07 (2.96–10.25)	12.84 (8.39–16.62)	2.10	0.04 ^α^
CC	14	21.48 (2.85–23.18)	15.19 (11.61–25.74)	1.57	0.12
** *BsmI* **					
GG	9	21.67 (15.53–24.07) ^╣^	17.30 (11.61–27.49)	1.12	0.26
GA	20	5.97 (2.94–12.91)	13.33 (8.93–20.16)	2.87	<0.01 ^α^
AA	6	5.69 (3.35–11.10) ^╣^	24.65 (14.95–28.50)	1.57	0.12
** *FokI* **					
TT	5	9.12 (3.62–12.78)	10.86 (4.75–37.76)	1.48	0.14
TC	15	16.67 (3.48–22.67)	14.16 (11.99–23.57)	1.31	0.19
CC	15	6.22 (3.15–21.15)	16.94 (9.95–26.41)	2.44	0.02 ^α^
**Total**	35	9.12 (3.22–21.48)	14.96 (10.97–25.74)	3.21	<0.01 ^α^

Data are presented as median (IQR); ^α^—Significant at *p* < 0.05; ^├^ and ^╣^—Significant at *p* < 0.05 during intra-seasonal analysis by Kruskal–Wallis ANOVA.

## Data Availability

The data presented in this study are available on request from the corresponding author. The data are not publicly available due to privacy restrictions.

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
