# Peer review of "Vitamin D Receptor Gene Polymorphism and Vitamin D Status in Population of Patients with Cardiovascular Disease—A Preliminary Study"

_nutrients, 2021, doi:10.3390/nu13093117_

Round 1

Reviewer 1 Report

The paper is interesting, but the presentation of data may be improved. The introduction is somewhat unclear and should be more directed to the potential significance of the polymorphisms and clinical outcomes or associations. The description of epimeric metabolites is very short, and the relevance of analyses of these metabolites are not clearly shown later in the paper. The method of vitamin D (UPCL-MS/MS) may be described a little more and consideration given to the choice of method. Table 1 may be improved with including distribution of tests during summer and winter. Furthermore, seasonal variation or distribution of tests is throughout the paper presented somewhat unclearly. I would recommend a separate paragraph explaining the significance of this regarding the results presented. It may be preferable to describe the vitamin D metabolites as 25(OH) D, 25(OH)D2, 25(OH)D3 etc. in the text (not vitD). The number of decimals on the p-values in tables and text are not consistent. There is no reason to have more than two decimals, and this should be corrected in text and tables. The discussion may be shortened, and the first few paragraphs belongs in the introduction. Moreover, the discussion should be started with a short summary of findings before relating these to other studies and considerations. Finally, a table or figure explaining the polymorphisms described would be useful for the general reader. All in all, however, the study is interesting even if it is rather small. 

Reviewer 2 Report

It is intriguing that the authors of this manuscript put forward the hypothesis that one or more of the polymorphic genotypes of the vitamin D receptor (VDR) could be associated with some aspect of cardiovascular disease. It is quite remarkable that significant associations were found between some of the genotypes and such conditions as hypertension, obesity, and hypercholesterolemia. Wisely, the authors have nominated two limitations of their study, namely the limited number of subjects that were investigated and that no mechanisms were apparent to explain the observed associations. Nevertheless, within the limitations of their study, the authors have collected data on every potential factor that might be involved, and especially the vitamin D status of the subjects.

There is one abbreviation that causes some confusion. That is the use of the term “vitD” as a catch-all for the overall vitamin D endocrine system. This term in some contexts could be interpreted as “vitamin D”. That is the parent compound, cholecalciferol. For example, at line 75, the term “20 ng/ml vitD” is used. Surely this refers to 25-hydroxyvitamin D. However, it could be interpreted as 20 ng/ml cholecalciferol. Likewise at lines 136 and 325 the abbreviation “vitD” is confusing. It would be preferable to use other abbreviations to avoid confusion.

Line 71: “vivo model” should read “in vivo model”.

Author Response

Reviewer: It is intriguing that the authors of this manuscript put forward the hypothesis that one or more of the polymorphic genotypes of the vitamin D receptor (VDR) could be associated with some aspect of cardiovascular disease. It is quite remarkable that significant associations were found between some of the genotypes and such conditions as hypertension, obesity, and hypercholesterolemia. Wisely, the authors have nominated two limitations of their study, namely the limited number of subjects that were investigated and that no mechanisms were apparent to explain the observed associations. Nevertheless, within the limitations of their study, the authors have collected data on every potential factor that might be involved, and especially the vitamin D status of the subjects.

1) There is one abbreviation that causes some confusion. That is the use of the term “vitD” as a catch-all for the overall vitamin D endocrine system. This term in some contexts could be interpreted as “vitamin D”. That is the parent compound, cholecalciferol. For example, at line 75, the term “20 ng/ml vitD” is used. Surely this refers to 25-hydroxyvitamin D. However, it could be interpreted as 20 ng/ml cholecalciferol. Likewise at lines 136 and 325 the abbreviation “vitD” is confusing. It would be preferable to use other abbreviations to avoid confusion.

 Response:

Thank you very much for your advice. We agree with you that the vitD abbreviation may cause confusion for the readers. Therefore, we have changed all the metabolites throughout the manuscript as follows:

  1. 25vitD to 25(OH)D
  2. 25vitD3 to 25(OH)D3
  3. 25vitD2 to 25(OH)D2
  4. 3-epi-vitD3 to epi-25(OH)D3
  5. 3-epi-vitD2 to epi-25(OH)D2
  6. Vitamin D3 to cholecalciferol
  7. Vitamin D2 to ergocalciferol
  8. vitD is dedicated only now to vitamin D

2) Line 71: “vivo model” should read “invivo model”.

Response:

 Thank you, we have corrected this to: "in vivo model"